# Effects of Different Irrigation Treatments on Aquaculture Purification and Soil Desalination of Paddy Fields

**Yi Xie [1], Zhenchang Wang [1],*, Xiangping Guo [1], Sirikanya Lakthan [1,2], Sheng Chen [1], Zhiming Xiao [1] and Yousef Alhaj Hamoud [1,3]**

[1]   College of Agricultural Engineering, Hohai University, Nanjing 210098, China
[2]   Royal Irrigation Department, Bangkok 10300, Thailand
[3]   Department of Soil and Land Reclamation, Aleppo University, Aleppo 1319, Syria
*   Correspondence: wangzhenchang@hhu.edu.cn

**Abstract:** Aquaculture wastewater contains considerable quantities of organic matter, nitrogen, and phosphorus. Irrigation of paddy rice with aquaculture wastewater can make full use of water and fertilizer, which has practical significance for alleviating water-use conflicts in the coastal areas of Jiangsu Province and promoting soil desalinization. Our objective in this study was to evaluate the effects of water quality indexes of surface discharge in paddy fields, total nitrogen (TN) and total phosphorus (TP) losses of discharge in paddy fields, growing indexes of plants, grain yield, as well as soil salinity affected by the different irrigation treatments. To achieve this objective, experiments were conducted from May to October in 2017. There were four treatment combinations: I1W1 (shallow–frequent irrigation and aquaculture wastewater), I2W1 (shallow–wet irrigation and aquaculture wastewater), I3W1 (flooding irrigation and aquaculture wastewater), and I1W2 (shallow–frequent irrigation and fresh water). The results revealed that there was no significant difference in grain yield among the three wastewater irrigation treatments. Meanwhile, the values of water quality indexes were optimal in I3W1; the total losses of TN and TP of the I3W1 treatment were the lowest over the three important growth stages; the desalinization rate of I3W1 was the largest due to its long hydraulic retention time and large irrigation depth. The overall results suggested that the I3W1 treatment was the optimal treatment.

**Keywords:** irrigation treatment; aquaculture purification; rice; growth; yield; soil desalination

## 1. Introduction

China feeds 21% of the world's population, while China's land resources account for 6.4% of the world [1]. Due to the sustained growth of land-use for industrial, environmental, and other purposes, as well as the increasing food demand caused by population growth, the shortage of land resources has become an urgent problem in China [2]. The area of tideland resources is approximately 3,540,000 ha in China, and is considered one of the most important agricultural land reserves, especially in eastern China [3]. Due to the high soil salinity in the early stage of reclamation, freshwater aquaculture is an effective mode for promoting soil desalinization [4]. In addition, reports have indicated that a wide range of plant species can grow in coastal saline areas such as salt marshes, and the establishment of plant communities on saline soil can alter the soil properties because of the biochemical and physical interactions between plants and soil [5]; studies have also shown that the establishment of plants in saline soil promotes soil desalinization [6]. Whether freshwater aquaculture or the establishment of plants, they both need a large amount of fresh water. China is a country lacking freshwater resources [7], and Jiangsu Province is an overloading type water deficit area [8].

Studies found that, in aquaculture, only about one-third of the nitrogen and phosphorus nutrients in the form of bait and fertilizer were absorbed by aquatic animals, and most of the remaining nutrients remained in the aquaculture wastewater and sediment in the form of residual baits and excreta [9]. The aquaculture wastewater discharged directly into the sea, increasing the pollution of coastal waters. According to Bulletin on the State of China's Marine Environment in 2011, the coastal areas of Jiangsu Province were severe eutrophication areas. Xiong and He found that, during the hot summer (July to August), the discharge depth of the aquaculture could reach 2000~4000 mm [10], and the nitrogen and phosphorus loads could reach 222 kg/hm$^2$ and 25 kg/hm$^2$, respectively [11]. The paddy rice growth period is from May to October in Jiangsu Province, and it is a water-loving crop and requires a large amount of water under flood-irrigated conditions [12], whereas, the local aquaculture discharge concentration is high and the discharge amount is large during this period. The discharge of aquaculture contains a large amount of nutrients such as nitrogen and phosphorus, which are easily absorbed by paddy rice. The nutrients are beneficial in increasing the tillering rate and number of panicles [13]. Through proper irrigation and discharge treatments, reuse of aquaculture wastewater for agricultural irrigation is undoubtedly a key strategy to reduce fresh water consumption [14]. Irrigation of paddy rice with aquaculture wastewater can make full use of water and fertilizer, which has practical significance for alleviating water use conflicts in the coastal areas of Jiangsu Province and promoting soil desalinization.

The aim of this study was to combine aquaculture purification with soil desalination on saline soil. The objectives were to determine (1) the changes in water quality indexes for aquaculture wastewater in different irrigation treatments; (2) the changes in TN and TP losses for discharge in different irrigation treatments; (3) the changes in growth indexes for paddy rice in different irrigation treatments; and (4) the effect of soil desalinization under aquaculture wastewater in different irrigation treatments.

## 2. Materials and Methods

### 2.1. Experimental Setup

The experiment was conducted from June to October 2017 at the Laboratory of Efficient Irrigation–Drainage and Agriculture Soil–Water Environment (31°86′N, 118°60′E), Hohai University, Nanjing, China. This area is characterized as a humid subtropical climate and is under the influence of the East Asia Monsoon, with an average annual rainfall of 1062 mm. The annual mean temperature is 15.5 °C with a monthly mean ranging from 2.4–27.8 °C; the highest recorded temperature in the area was 43.0 °C while the lowest was −16.9 °C.

The soil was taken from the coastal area of Dongtai, Jiangsu Province. The soil sample was air-dried, ground, and passed through a 1 mm sieve. The EC, soluble cations, and anions in the soil were analyzed in 1:5 soil water extracts. The Cl$^-$ content was determined by titration with AgNO$_3$, the SO$_4{}^{2-}$ content determined by EDTA indirect complexometric titration, the HCO$_3{}^-$ content was determined by the double-indicator neutral method, and the Na$^+$, K$^+$, Ca$^{2+}$, and Mg$^{2+}$ contents were measured using an atomic absorption spectrophotometer (UV1901, Shanghai, China) [15]. The soil was classified as silty sand with an EC1:5 of 4.1 mS cm$^{-1}$. The concentrations of Ca$^{2+}$, Mg$^{2+}$, Na$^+$, K$^+$, CO$_3{}^{2-}$, HCO$_3{}^-$, Cl$^-$, and SO$_4{}^{2-}$ in the saline soil with the EC 1:5 value of 4.1 mS cm$^{-1}$ were 2.58 mmol kg$^{-1}$, 0.82 mmol kg$^{-1}$, 70.02 mmol kg$^{-1}$, 0.21 mmol kg$^{-1}$, 2.32 mmol kg$^{-1}$, 9.21 mmol kg$^{-1}$, 84.31 mmol kg$^{-1}$, and 3.58 mmol kg$^{-1}$, respectively.

### 2.2. Experimental Design and Irrigation Management

The shelter experiment was conducted using randomized complete blockdesign (RCBD) with a factorial arrangement of treatments with four replications under nature light conditions without temperature control. The experiment comprised three irrigation treatments (I), namely, shallow–frequent irrigation (I1), shallow–wet irrigation (I2), and flooding irrigation (I3), and two sources of irrigation water (W), namely, aquaculture wastewater (W1) and fresh water (W2). In consideration

of the quantity and frequency of exchanged water in the aquiculture pond [16], as well as purification effects for aquaculture wastewater in the paddy field [17], high irrigation frequency and high irrigation depth treatments were introduced in this study. Thus, there were four treatment combinations: I1W1 (I1 and W1), I2W1 (I2 and W1), I3W1 (I3 and W1), and I1W2 (I1 and W2). A randomized complete block design with all four treatments and four replicates was established in a plot of approximately 20 m². The plastic boxes used for this study were 90 cm long and 68 cm wide and 67 cm deep.

Before the experiment, the PVC tube (having a 2 cm inner diameter and 70 cm in length) was installed vertically at a depth of 60 cm in every plastic box to collect percolation water. The cruciform PVC tube with numerous pores (approximately 2 μm in diameter) was laid on the bottom of the plastic box, connected to the vertical PVC tube, and then a 7 cm gravel filter layer was laid on it. The cruciform PVC tubes were surrounded by nonwovens to prevent soil particles plugging the pores. After, air-drying and sieving, the soil was compacted (every 10 cm compaction) into the box, and each box was reserved for 20 cm water storage depth. The percolation water was sampled at 3 day intervals during the entire growing season with a vacuum pump.

The variety of rice planted in this area was Japonica Rice Nanjing 5055. The paddy rice seeds were sown on 10 May for the 2017 season. When the third true leaf of the seedling had grown (21 June for the 2017 season), seedlings of similar height were selected to transplant to the plastic boxes. The plants were spaced at 20 × 15 cm per box and three seedlings per hole. There were 12 holes of rice per box. The entire growth season was divided into six stages: re-greening stage, tillering stage, jointing and booting stage, heading and flowering stage, milk maturity stage, and yellow maturity stage. The paddy rice was harvested on 27 October 2017. Water table depth controls and the duration of flooding in different stages for three irrigation treatments were shown in Table 1. The fertilization process conducted in this experiment were shown in Table 2. Only the basal fertilizers were incorporated into the ploughed layer, the other fertilizers were broadcast onto the soil surface. The water depths were measured using vertical rulers. Pest and weed control management in this experiment followed local rice cultivation practices.

**Table 1.** Water table depth control and the duration of flooding in different stages for three irrigation treatments.

| Irrigation Treatments | Limit | Re-Greening | Tillering | | Jointing and Booting | Heading and Flowering | Milk Maturity | Yellow Maturity |
|---|---|---|---|---|---|---|---|---|
| | | | Initial | Late | | | | |
| I1 [1] | Upper limit of irrigation/mm | 20~30 | 15 | 35 | 50 | 50 | 50 | 0 |
| | The duration of aquaculture wastewater kept in the soil surface/d | | 1 | 1 | 1 | 1 | 1 | Naturally drying |
| I2 | Upper limit of irrigation/mm | 20~30 | 25 | 50 | 100 | 100 | 100 | 0 |
| | The duration of aquaculture wastewater kept in the soil surface/d | | 2 | 2 | 2 | 2 | 2 | Naturally drying |
| I3 | Upper limit of irrigation/mm | 20~30 | 35 | 65 | 150 | 150 | 150 | 0 |
| | The duration of aquaculture wastewater kept in the soil surface/d | | 3 | 3 | 3 | 3 | 3 | Naturally drying |

[1] I1 is shallow–frequent irrigation; I2 is shallow–wet irrigation; I3 is flooding irrigation.

**Table 2.** Time and amount of fertilization (kg hm$^{-2}$).

| Year | Activity | W1 [2] | W2 |
|---|---|---|---|
| 2017 | Base fertilizer (21 June) [1] | 300.0 (CF) [3] | 300.0 (CF) |
| | Re-greening fertilizer (4 July) | 75.0 (U) | 150.0 (U) |
| | Tillering fertilizer (15 July) | 62.5 (U) | 125.0 (U) |
| | Panicle fertilizer (18 August) | 75.0 (U) | 150.0 (U) |
| | Total nitrogen | 512.5 | 725 |

[1] Date in the bracket is the time for the fertilizer applied. [2] W1 means that the source of irrigation water is aquaculture wastewater; W2 means that the source of irrigation water is fresh water. [3] CF is compound fertilizer (N, $P_2O_5$, and $K_2O$ contents were 15%, 15%, and 15% in 2016 and 2017). U is urea (N content was 46.2%).

## 2.3. Measurements

The plant height of the marked three-hole plants per box were continuously monitored. Plant height of each marked plant was measured using a 100 cm stainless steel ruler at 5 day intervals. The height from the base to the highest leaf tip was measured before the heading stage and the height from the base to the highest panicle after the heading stage. Tiller number of each plant was measured at 5 day intervals. Leaf area of each plant was measured at 5 day intervals with a portable laser leaf area meter (CI-203, CID Inc., Washington, DC, USA). Stem diameter of each plant was measured at 5 day intervals with a Shenhan electronic vernier caliper and a 100 cm stainless steel ruler. Measurements were made at a distance of 5 cm from the topsoil.

The experiment comprised three irrigation treatments (I), namely, shallow–frequent irrigation (I1), shallow–wet irrigation (I2), and flooding irrigation (I3). The duration of irrigation water kept in the soil surface was one day in I1; the duration of irrigation water kept in the soil surface was two day in I2; the duration of irrigation water kept in the soil surface was three day in I3. According to the three different irrigation treatments, we collected the water samples daily for I1, at 2 day intervals for I2, and at 3 day intervals for I3. Therefore, the surface water samples were collected and replaced with aquaculture wastewater daily for I1W1; the surface water samples were collected and replaced with aquaculture wastewater at 2 day intervals for I2W1; the surface water samples were collected and replaced with aquaculture wastewater at 3 day intervals for I3W1; the surface water samples were collected and replaced with fresh water daily for I1W2. According to the amount of deep percolation in local paddy field, we collected the ground water samples at 3 day intervals for all treatments. At each time of discharge, water samples were collected from each plastic box by pump. Additionally, water samples were collected in 250 mL bottles and packed on ice for transportation to the laboratory for chemical analysis. Filtered samples were analyzed for water quality indexes including dissolved oxygen (DO), electrical conductivity (EC), potential of hydrogen (pH), and oxidation-reduction potential (ORP). These water quality indexes were measured at the experiment site at the time of sampling using the Hydrolab DS5X (Hach, Loveland, CO, USA). We monitored water quality continuously during the two key growing stages (i.e., tillering stage and jointing and booting stage), because this period was important for rice growth and the water in aquaculture were replaced frequently due to the high temperature during this period. Filtered samples were analyzed for total nitrogen (TN) and total phosphorus (TP). Total nitrogen in the water samples was determined using alkaline potassium persulfate digestion and the UV spectrophotometric method, respectively. Total phosphorus measured as $PO4^{3-}$ following persulfate digestion was quantified using the ammonium molybdate spectroscopic method (MDL = 0.005 mg L$^{-1}$) [18]. In this study, during the three key growing stages (i.e., tillering stage, jointing and booting stage, heading and flowering stage), a continuous period of time was selected for the determination of TN and TP.

The grain yield is the total dry weight of the rice grains. The rice quality was measured on harvesting stage. The plant samples were divided into leaves, stems, and rice panicles, and the samples were dried at 105 °C in the oven for 15 to 30 min, respectively. Then, the samples were dried at 75 °C in the oven until a constant weight was achieved. After, the dry biomass of plant samples was weighed. The length of effective heads, the number of effective heads, number of grains per head, seed setting rate, and the weight of one thousand grains were measured.

At each sampling point, soil samples were collected. Soil electrical conductivity (EC) measurements were conducted in situ with a portable and corrected sensor probe (Spectrum Technologies Inc., Texas, TX, USA). Within the same core, when the soil samples were taken out with an auger, the sensor probes were inserted into the soil to measure the electrical conductivity, and each layer was measured three times. The average value was calculated as the representative value of the EC for the layer.

## 2.4. Statistical Analysis

One-way analysis of variance (ANOVA) was performed using the general linear model-univariate procedure with SPSS 13.0 software (SPSS, Chicago, IL, USA). When significant differences were detected,

mean values of each treatment were compared using Duncan's multiple range tests. Unless otherwise stated, the significance level was $p \leq 0.05$.

## 3. Results

### 3.1. Water Quality Indexes

The dynamics of water quality indexes for surface discharge in paddy fields of different irrigation treatments is shown in Figure 1.

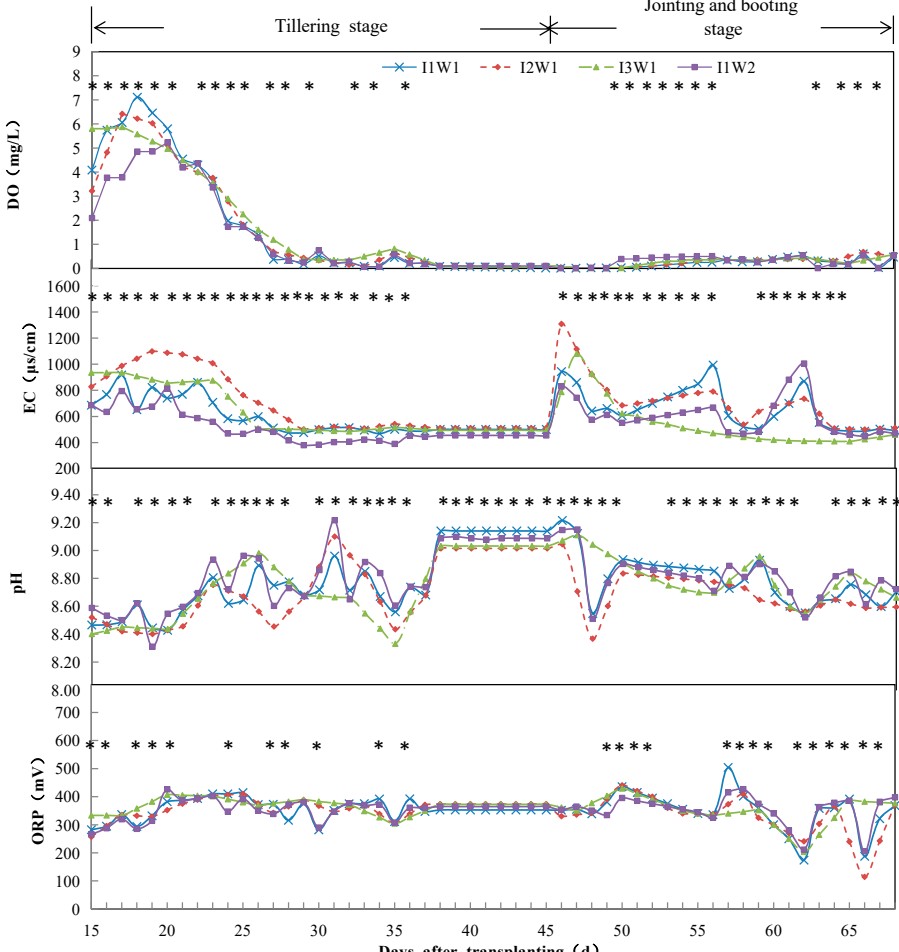

**Figure 1.** Temporal variations over the growing season in water quality indexes for surface discharge in different irrigation treatments. DAT denotes days after transplanting; DO means dissolved oxygen; EC means electrical conductivity; pH means potential of hydrogen; ORP means oxidation-reduction potential. mg/L means milligrams per liter; μS/cm means microsecond per centimeter; mV means millivolt. I1W1 represents shallow–frequent irrigation with aquaculture wastewater; I2W1 represents shallow–wet irrigation with aquaculture wastewater; I3W1 represents flooding irrigation with aquaculture wastewater; I1W2 represents shallow–frequent irrigation with fresh water. Asterisks indicate where means of one treatment EC value was significantly different from another EC value (Duncan test, $p \leq 0.05$).

### 3.1.1. Temporal Variation of DO

The broad trend of DO for surface discharge was downward over the growing stage and it decreased sharply in the late tillering stage. The DO variations were similar in I1W1, I2W1, I3W1, and I1W2 over the tillering stage. Dissolved oxygen (DO) increased at 15 DAT then decreased during the tillering stage. For surface discharge, DO decreased from 7.11 mg/L to 0.068 mg/L with the highest value

for the I1W1 treatment and the lowest value for the I1W2 treatment in the tillering stage. Compared with the I1W1 treatment, the peak DO values (7.11 mg/L, 6.42 mg/L, 5.88 mg/L, and 5.23 mg/L for I1W1, I2W1, I3W1, and I1W2) decreased by 9.70%, 17.37%, and 26.48% for I2W1, I3W1, and I1W2 treatments over the initial tillering stage, respectively; compared with I3W1 treatment, the peak DO values (0.45 mg/L, 0.64 mg/L, 0.81 mg/L, and 0.58 mg/L for I1W1, I2W1, I3W1, and I1W2) decreased by 44.18%, 21.93%, and 28.38% at 35 DAT in I1W1, I2W1, and I1W2 treatments over the late tillering stage, respectively. During the late tillering stage, there were no differences in water quality indexes among all treatments because we did not irrigate and drain water during the period. In addition, we did not collect water samples during the period. The DO variations were similar in I1W1, I2W1, I3W1, and I1W2 in the jointing and booting stage, which varied from 0.01 mg/L to 0.068 mg/L.

### 3.1.2. Temporal Variation of EC

For surface discharge, EC ranged from 1097.95 μS/cm to 384.38 μS/cm, with the highest value for I2W1 treatment and the lowest value for I1W2 treatment in the tillering stage. In I1W1, the EC increased first, then varied widely from 18 DAT to 28 DAT, and was maintained at approximately 480 μS/cm thereafter over the tillering stage; in I2W1, EC first increased, then decreased at 19 DAT, and was maintained at approximately 510 μS/cm thereafter; in I3W1, the EC was maintained at approximately 900 μS/cm from 15 DAT to 23 DAT, then decreased to 500 μS/cm at 28 DAT, and was maintained at approximately 500 μS/cm thereafter; in I1W2, the EC variation was similar to the I1W1, and was maintained at approximately 405 μS/cm at the end. However, there was an uptrend in the EC concentrations in the initial jointing and booting stage. During the jointing and booting stage, EC ranged from 1308.25 μS/cm to 409.35 μS/cm with the highest values at I2W1 and the lowest values at I3W1 for surface discharge. The EC variations were similar in I1W1, I2W1, and I1W2. The EC decreased from the peak values (approximately 942.83 μS/cm, 1308.25 μS/cm, and 830.27 μS/cm for I1W1, I2W1, and I1W2) at 46 DAT to the low values (approximately 606.03 μS/cm, 684.83 μS/cm, and 551.33 μS/cm for I1W1, I2W1, and I1W2) at 50 DAT, then EC varied widely from 51 DAT to 64 DAT, and was maintained at the stable values (approximately 490 μS/cm, 505 μS/cm, and 470 μS/cm for I1W1, I2W1 and I1W2) thereafter over the jointing and booting stage; in I3W1, EC decreased from approximately 1080 μS/cm at 47 DAT to 409.35 μS/cm at 65 DAT.

### 3.1.3. Temporal Variation of pH

The pH varied widely in the tillering stage. For surface discharge, pH ranged from 8.31 to 9.22 with the highest value at 31 DAT in the I1W2 treatment and the lowest value at 19 DAT in the I1W2 treatment over tillering stage. In I1W1, the pH ranged from 8.43 to 9.14 with the lowest value at 20 DAT and the highest value at 38 DAT; in I2W1, the pH ranged from 8.40 to 9.10 with the lowest value at 19 DAT and the highest value at 31 DAT; in I3W1, the pH first increased, then pH decreased at 26 DAT, and the pH increased to 9.02 at 38 DAT; in I1W2, the pH ranged from 8.31 to 9.22 with the lowest value at 19 DAT and the highest value at 31 DAT. During the jointing and booting stage, pH ranged from 8.37 to 9.22 with the highest value at I1W1 and the lowest value at I2W1 for surface discharge. The pH variations were similar in I1W1, I2W1, and I1W2, and the pH decreased from the peak value at 46 DAT to the low value at 48 DAT, then pH increased at 50 DAT and decreased, then varied widely with the lowest value at 62 DAT; in I3W1, the pH first decreased, then increased to 8.96 at 59 DAT, and varied widely from 59 DAT to 68 DAT.

### 3.1.4. Temporal Variation of ORP

The ORP varied widely in the tillering stage. For surface discharge ORP ranged from 257 mV to 426 mV with the lowest value at 15 DAT in the I2W1 treatment and the highest value at 20 DAT in the I1W2 treatment in the tillering stage. Generally, the ORP first increased and then decreased in the tillering stage. During the late tillering stage, there were no differences in water quality indexes among all treatments. We did not collect water samples during this period. The ORP varied widely in

the jointing and booting stage. For surface discharge, ORP ranged from 115 mV to 504 mV with the lowest value at 66 DAT in I2W1 treatment and the highest value at 57 DAT in the I1W1 treatment in the jointing and booting stage.

### 3.2. The TN and TP Losses through Discharge

The TN and TP losses for surface discharge in different irrigation treatments are shown in Figures 2 and 3. The TN and TP losses for ground discharge in different irrigation treatments are shown in Figures 4 and 5.

### 3.2.1. The TN Losses for Surface Discharge

Total nitrogen concentrations and losses in the tillering stage were higher than other stages in I1W1, I2W1, and I1W2 treatments. Total nitrogen concentrations and losses were the lowest in the heading and flowering stage. Total nitrogen concentration in the I2W1 treatment was the largest in the tillering stage, followed by I1W2 treatment. Compared with the I2W1 and I1W2 treatments, the concentration of TN in the I1W1 treatment significantly decreased. Total nitrogen concentration in the I1W1 treatment showed a significantly higher value than the I3W1 treatment ($p \leq 0.05$, Figure 2). Significant differences in the TN concentrations among the treatments were found during the jointing and booting stage. Total nitrogen concentration in the I1W2 treatment was the highest, followed by I3W1 and I2W1. Total nitrogen concentration in I1W1 was the lowest ($p \leq 0.05$, Figure 2). In the heading and flowering stage, the mean concentrations of TN in all treatments were not significantly different. In the three key growing stages, the mean concentration of TN in the I1W2 treatment was the highest, followed by I2W1 and I1W1, whereas TN concentration in I3W1 was the lowest ($p \leq 0.05$, Figure 2).

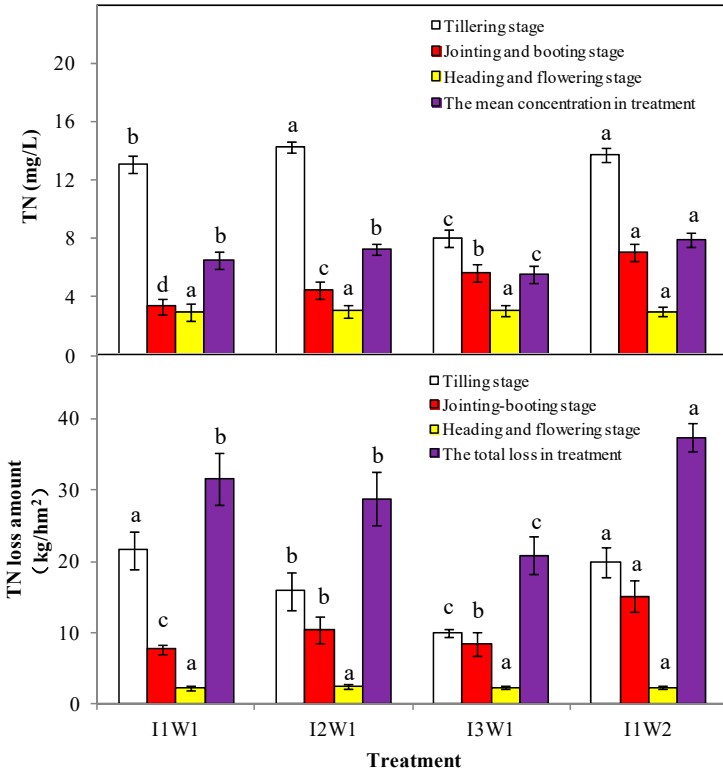

**Figure 2.** Total nitrogen (TN) losses for surface discharge in different irrigation treatments. I1W1 represents shallow–frequent irrigation with aquaculture wastewater; I2W1 represents shallow–wet irrigation with aquaculture wastewater; I3W1 represents flooding irrigation with aquaculture wastewater; I1W2 represents shallow–frequent irrigation with fresh water. Data are mean ± SD ($n$ = 4). The different letters on the tops of columns indicate significant differences among treatments at 5% according to Duncan's multiple range test.

The losses of TN were significantly affected by the mean concentrations of TN for surface discharge. Total nitrogen losses in the I1W1 treatment were the largest in the tillering stage, followed by I1W2. Comparing the I1W1 and I1W2 treatments, the losses of TN in the I2W1 treatment significantly decreased. Total nitrogen losses in the I2W1 treatment showed a significantly higher value than I3W1 ($p \leq 0.05$, Figure 2). Total nitrogen losses in the I1W1 treatment were the largest in the jointing and booting stage, followed by I3W1 and I2W1. While comparing with the I1W1 treatment, the losses of TN in the I3W1 and I2W1 treatments significantly decreased. Total nitrogen losses in the I1W1 treatment were the lowest and showed a significantly lower value than the other three treatments ($p \leq 0.05$, Figure 2). There were no differences among all the treatments for TN losses in the heading and flowering stage. The variations of the losses of TN in the three key growing stages were the same as the variations of mean concentrations of TN.

3.2.2. The TP Losses for Surface Discharge

Total phosphorus concentrations and losses in all four treatments were the highest in the tillering stage, followed by those in the jointing and booting stage. Total phosphorus concentrations and losses were the lowest in the heading and flowering stage. Total phosphorus concentration in the I3W1 treatment was the largest in the tillering stage, followed by I2W1 and I1W2, respectively. When comparing with the sI3W1 treatment, the concentrations of TP in the I2W1 and I1W2 treatments significantly decreased. Total phosphorus concentration in I1W1 showed a significantly lower value than I2W1 and I1W2 ($p \leq$ 0.05, Figure 3). Total phosphorus concentration in I1W2 was the largest in the jointing and booting stage, followed by I3W1. While comparing with the I1W2 treatment, the concentrations of TP in the I1W1 and I2W1 treatments significantly decreased. Total phosphorus concentrations in I1W1, I2W1, and I3W1 were not significantly different ($p \leq 0.05$, Figure 3). Total phosphorus concentration in the I1W1 treatment was the largest in the heading and flowering stage, followed by I1W2 and I3W1. Total phosphorus concentration in the I2W1 treatment was the lowest. Comparing with the I1W1 treatment, the concentration of TP in the I1W2 treatment significantly decreased. Comparing with the I1W2 treatment, the concentrations of TP in the I2W1 and I3W1 treatments significantly decreased. In the three key growing stages, the mean concentration of TP in I3W1 was the largest, followed by I2W1 and I1W2, respectively, whereas TP concentration of I1W1 was the lowest ($p \leq 0.05$, Figure 3).

Total phosphorus losses in all four treatments in the tillering stage were higher than those in the jointing and booting stage and the heading and flowering stage. So, the TP losses in the tillering stage were vital. Total phosphorus losses in the I1W2 treatment were the largest in the tillering stage, followed by I1W1 and I2W1. Comparing with the I1W2 treatment, the losses of TP in the I1W1 and I2W1 treatments significantly decreased in the tillering stage. Total phosphorus losses in I3W1 showed a significantly lower value than I1W1 and I2W1 ($p \leq 0.05$, Figure 3). Total phosphorus losses in I1W2 were the largest in the jointing and booting stage, followed by I1W1, I2W1, and I3W1. While comparing with the I1W2 treatment, the losses of TP in I1W1, I2W1, and I3W1 treatments significantly decreased ($p \leq 0.05$, Figure 3). Total phosphorus losses in I1W1 were the largest in the heading and flowering stage, followed by I1W2 and I3W1, whereas TP losses in I2W1 were the lowest.

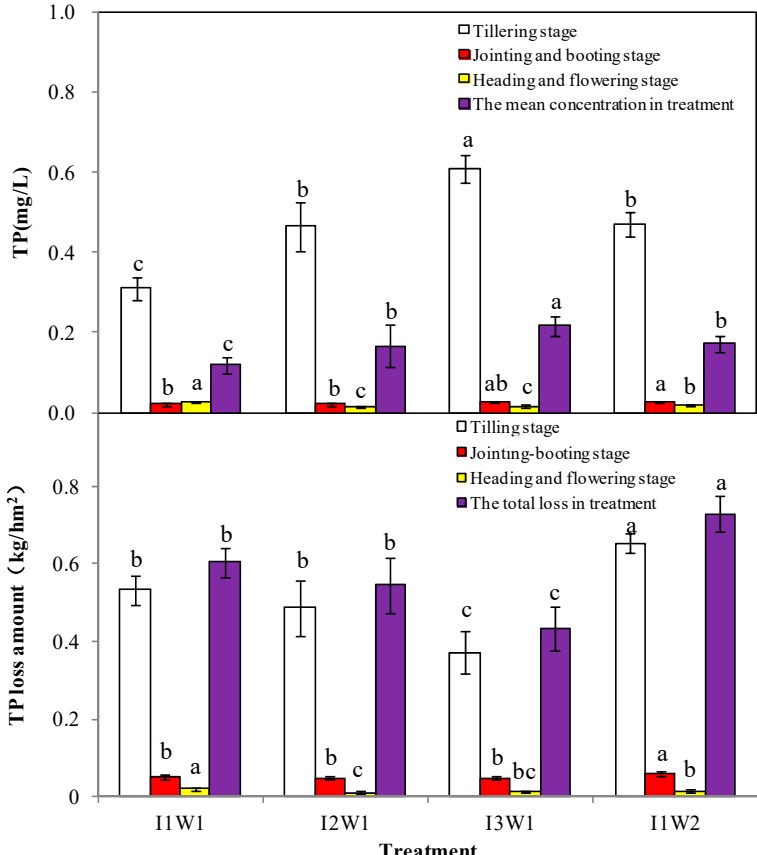

**Figure 3.** Total phosphorous (TP) losses for surface discharge in different irrigation treatments. I1W1 represents shallow–frequent irrigation with aquaculture wastewater; I2W1 represents shallow–wet irrigation with aquaculture wastewater; I3W1 represents flooding irrigation with aquaculture wastewater; I1W2 represents shallow–frequent irrigation with fresh water. Data are mean ± SD (*n* = 4). The different letters on the tops of columns indicate significant differences among treatments at 5% according to Duncan's multiple range test.

### 3.2.3. The TN Losses for Ground Discharge

Total nitrogen concentrations and losses in all four treatments in the tillering stage were higher than those in the jointing and booting stage and the heading and flowering stage. Total nitrogen concentration in the I3W1 treatment was the largest in the tillering stage, followed by I2W1 and I1W1. Total nitrogen concentration in I1W2 was the lowest. Comparing with the I3W1 and I2W1 treatments, TN concentrations in the I1W1 and I1W2 treatments significantly decreased ($p \leq 0.05$, Figure 4). Total nitrogen concentration in I1W2 was the largest in the jointing and booting stage, followed by I3W1 and I1W1. Total nitrogen concentration of I2W1 was the lowest. Total nitrogen concentrations in the I1W1 and I2W1 treatments significantly decreased when compared with the I1W2 and I3W1 treatments ($p \leq 0.05$, Figure 4). Total nitrogen concentration in I1W1 was the largest in the heading and flowering stage, followed by I2W1 and I1W2. Total nitrogen concentration in I3W1 was the lowest.

Total nitrogen losses in I3W1 treatment were the largest in the tillering stage, followed by I1W2 and I2W1 respectively, whereas TN losses in I1W1 were the lowest. Total nitrogen losses were significantly different among each treatment except I1W1 ($p \leq 0.05$, Figure 4). Total nitrogen losses in I1W2 were the largest in the jointing and booting stage, followed by I3W1 and I1W1, and TN losses in I2W1 were the lowest. Total nitrogen losses were significantly different among each treatment except I2W1, however, there was no significant difference between I1W1 and I2W1 in heading and flowering stage ($p \leq 0.05$, Figure 4). Total nitrogen losses in I1W2 were the largest in the heading and flowering stage, followed

by I2W1 and I1W1, and TN losses in I3W1 were the lowest. Total nitrogen losses were significantly different among each treatment except I3W1 ($p \leq 0.05$, Figure 4).

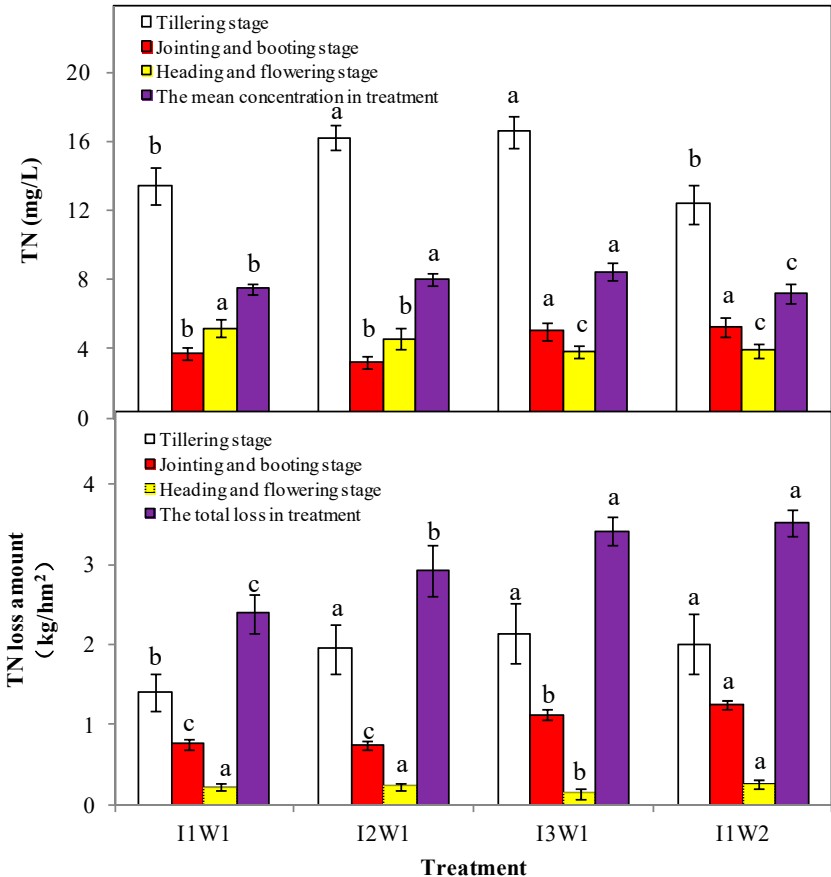

**Figure 4.** TN losses for ground water discharge in different irrigation treatments. TN means total nitrogen. I1W1 represents shallow–frequent irrigation with aquaculture wastewater; I2W1 represents shallow–wet irrigation with aquaculture wastewater; I3W1 represents flooding irrigation with aquaculture wastewater; I1W2 represents shallow–frequent irrigation with fresh water. Data are mean ± SD (*n* = 4). The different letters on the tops of columns indicate significant differences among treatments at 5% according to Duncan's multiple range test.

3.2.4. The TP Losses for Ground Discharge

Total phosphorus concentration in I1W1 was the largest in the tillering stage, followed by I1W2, I3W1, and I2W1. The I1W1 showed a significantly higher value than other treatments. The TP concentration of water samples was not significantly different among each treatment except I1W1 during the tillering stage ($p \leq 0.05$, Figure 5). The TP losses in I1W2 were the largest in the tillering stage, followed by I1W1 and I3W1, and the TP losses in I2W1 were the lowest. The TP losses of water samples were not significantly different among each treatment except I1W2 ($p \leq 0.05$, Figure 5). There was no significant difference among all four treatments of TP concentrations over the jointing and booting stage ($p \leq 0.05$, Figure 5). Total phosphorus losses in I1W2 were the largest over the jointing and booting stage, followed by I3W1 and I1W1, and TP losses in I2W1 were the lowest. Comparing with the I1W2 and I3W1 treatments, TP losses in the I1W1 and I2W1 treatments significantly decreased in the jointing and booting stage ($p \leq 0.05$, Figure 5). However, there was no significant difference among all four treatments of TP concentrations and losses over the heading and flowering stage ($p \leq 0.05$, Figure 5). Over the three key growing stages, the total losses of TP in the I1W2 treatment were the largest, followed by I1W1, I2W1, and I3W1 ($p \leq 0.05$, Figure 5).

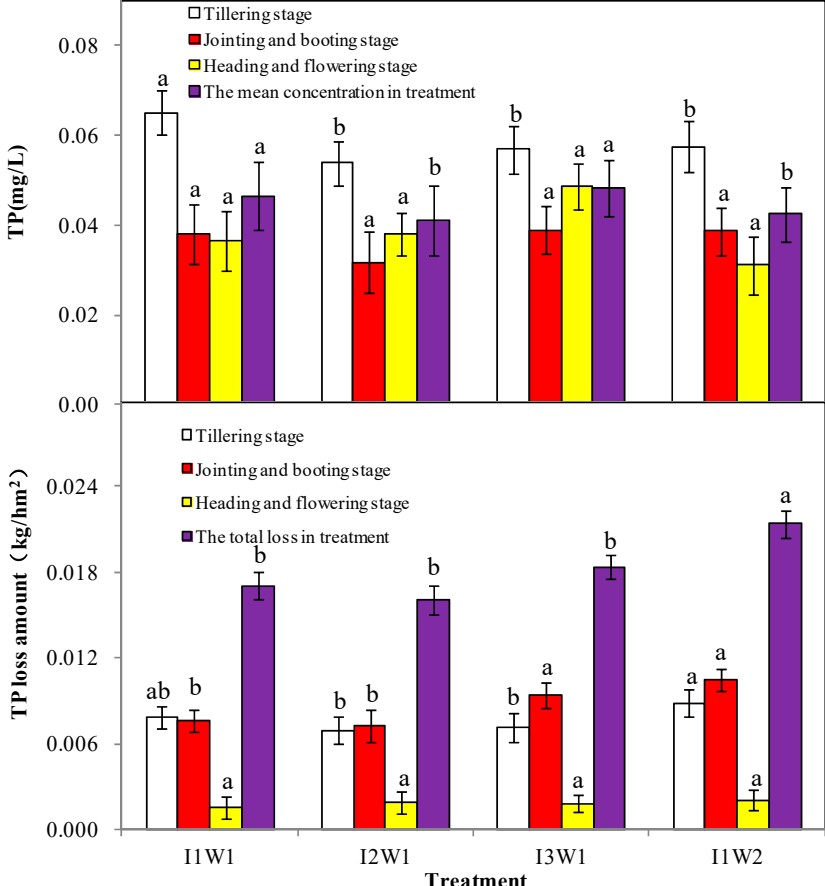

**Figure 5.** TP losses for ground water discharge in different irrigation treatments. TP means total phosphorus. I1W1 represents shallow–frequent irrigation with aquaculture wastewater; I2W1 represents shallow–wet irrigation with aquaculture wastewater; I3W1 represents flooding irrigation with aquaculture wastewater; I1W2 represents shallow–frequent irrigation with fresh water. Data are mean ± SD (*n* = 4). The different letters on the tops of columns indicate significant differences between treatments at 5% according to Duncan's multiple range test.

## 3.3. The Change of Growth Indexes

The dynamics of growth indexes in different irrigation treatments are shown in Figure 6.

### 3.3.1. Temporal Variation of Plant Height

In the tillering stage, the plant height variation in I2W1 treatment was similar to the I3W1. The plant height in I1W1 was lower than the other treatments over the tillering stage. In contrast, the plant height in I1W2 was higher than the other treatments and the plants in I1W2 grew faster than other treatments over the tillering stage. Based on the data in Figure 6, the plant height in I1W2 at the end of tillering stage was the highest (53.82 cm), followed by that in I2W1 (51.49 cm) and I3W1 (50.58 cm), and I1W1 had the lowest value (45.89 cm); compared with the I1W2 treatment, the plant height in I1W1, I2W1, and I3W1 treatments decreased by 19.99%, 9.37%, and 10.32% at the end of the tillering stage, respectively. The rate of growth during the jointing and booting stage was faster than other growing stages in all treatments. Based on the data in Figure 6, the rate of growth in I1W1 during the jointing and booting stage was highest (47.72%), followed by that in I2W1 (37.29%) and I3W1 (35.04%), and I1W2 had the lowest value (24.55%). The difference in plant height was reduced in all treatments during the jointing and booting stage. The plant height in I1W2 at the end of the jointing and booting stage was the highest, followed by that in I2W1 and I3W1, and I1W1 had the lowest value; compared with the I1W2 treatment, the plant height in the I1W1, I2W1, and I3W1 treatments decreased

by 3.70 cm, 0.08 cm, and 2.00 cm at the end of the booting stage, respectively. The plant heights in all four treatments during the tillering stage and jointing and booting stage showed significant values with each other ($p \leq 0.05$, Figure 6). The rates of growth during the heading and flowering stage and milk maturity stage were slow in all treatments, and the rates of growth were 6.06%, 0.83%, 1.80%, and 0.92% in I1W1, I2W1, I3W1, and I1W2, respectively. The plant height in I1W2 at the end of milk stage was the highest (73.09 cm), followed by that in I2W1 (72.95 cm) and I1W1 (72.89 cm), and I3W1 had the lowest value (71.69 cm); compared with I1W2 treatment, the plant height in I1W1, I2W1, and I3W1 treatments decreased by 0.27%, 0.19%, and 1.95% at the end of the milk stage, respectively.

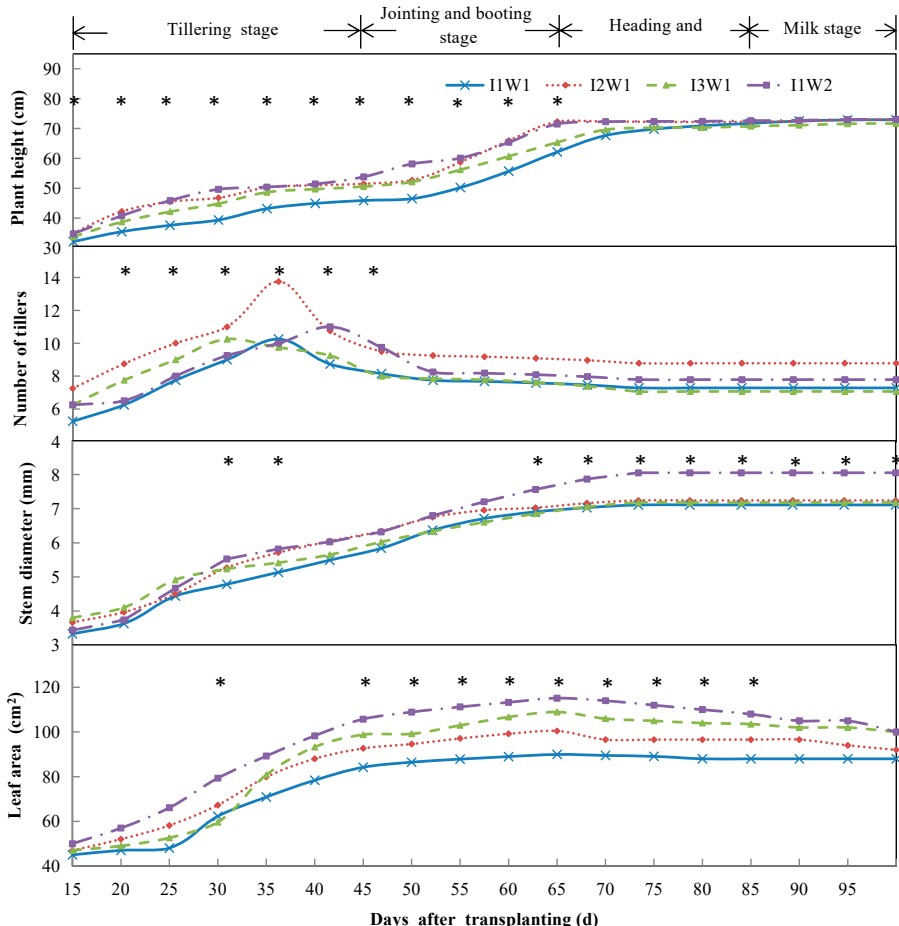

**Figure 6.** Temporal variation of growth parameters in different irrigation treatments. DAT denotes days after transplanting. I1W1 represents shallow–frequent irrigation with aquaculture wastewater; I2W1 represents shallow–wet irrigation with aquaculture wastewater; I3W1 represents flooding irrigation with aquaculture wastewater; I1W2 represents shallow–frequent irrigation with fresh water. Asterisks indicate where means of one treatment growth index was significantly different from another growth index (Duncan test, $p \leq 0.05$).

### 3.3.2. Temporal Variation in the Number of Tillers

The number of tillers in the four treatments showed similar variations in the tillering stage. In the tillering stage, the number of tillers first increased to 10.25, 13.75, 10.25, and 11.00, respectively, then the number of tillers decreased to 8.15, 9.50, 8.00, and 9.75 at 45 DAT, respectively. Based on the data in Figure 6, considering the number of tillers during the growing stage, I3W1 was the earliest to reach the peak (30 DAT), followed by that in I1W1 (35 DAT) and I2W1 (35 DAT), and I1W2 was the latest (40 DAT). At the end of the tillering stage, the numbers of tillers were 7.25, 8.75, 7.00, and 7.75 in the I1W1, I2W1, I3W1, and I1W2 treatments, respectively. Then the numbers of tillers were maintained thereafter over the following stage.

### 3.3.3. Temporal Variation in Stem Diameters

In the tillering and jointing and booting stages, the stem diameters continued growing. In the tillering stage, the stem diameter increased from approximately 3.50 mm to 5.84 mm, 6.34 mm, 6.03 mm, and 6.32 mm at 45 DAT in I1W1, I2W1, I3W1, and I1W2, respectively. Then, the stem diameter increased to 7.03 mm, 7.16 mm, 7.05 mm, and 7.87 mm in the jointing and booting stage, respectively, and was maintained thereafter over the following stage. Based on the data in Figure 6, the stem diameter was the highest in I2W1 at the end of the tillering stage, followed by I1W2 and I3W1, and I1W1 had the lowest value; comparing with the I2W1 treatment, the stem diameter in the I1W1, I3W1, and I1W2 treatments decreased by 0.50 mm, 0.32 mm, and 0.03 mm, respectively. The rates of growth were 44.00%, 40.48%, 38.11%, and 46.81% in I1W1, I2W1, I3W1, and I1W2 in the tillering stage, respectively. The nutritive growth of plants continued in the jointing and booting stage. The stem diameter in I1W2 was the highest in I1W2 at the end of jointing and booting stage, followed by I2W1 and I3W1, and I1W1 had the lowest value; comparing with the I1W2 treatment, the stem diameter in the I2W1, I3W1, and I1W1 treatments decreased by 10.06%, 10.82%, and 11.75%, respectively; comparing with the I2W1 treatment, the stem diameter in the I3W1 and I1W1 treatments decreased by 0.84% and 1.87%, respectively.

### 3.3.4. Temporal Variation of Leaf Area

In the tillering and jointing and booting stages, the leaf area continued growing. In the tillering stage, the leaf area increased from approximately 48.00 cm$^2$ to 84.21 cm$^2$, 92.66 cm$^2$, 98.79 cm$^2$, and 105.75 cm$^2$ at 45 DAT in I1W1, I2W1, I3W1, and I1W2, respectively. Then the leaf area increased to 89.96 cm$^2$, 100.43 cm$^2$, 108.94 cm$^2$, and 115.12 cm$^2$ in the jointing and booting stage, respectively. Based on the data in Figure 6, the leaf area was the highest in I1W2 at the end of the tillering stage, followed by I3W1 and I2W1, and I1W1 had the lowest value; the leaf area was the highest in I1W2 at the end of jointing and booting stage, followed by I3W1 and I2W1, and I1W1 had the lowest value. The leaf area continued decreasing in the heading and flowering and milk stages; the leaf area was the highest in I1W2 at the end of the milk stage, followed by I1W1 and I3W1, and I2W1 had the lowest value.

### 3.4. Grain Yields and Soil Salinity

Table 3 shows grain yields in different irrigation treatments. There was no significant difference in the number of grains per panicle, seed setting rate, and thousand grain weight among all four treatments ($p \leq 0.05$, Table 3). The grain yield in I1W2 was the highest, followed by that in I2W1 and I1W1, and that in I3W1 was the lowest. Comparing with the I1W2 treatment, the grain yields in the I1W1 and I3W1 treatments were significantly decreased; however, there was no difference between I1W2 and I2W1 ($p \leq 0.05$, Table 3). Table 4 shows soil salinity in different irrigation treatments. At the end of the experiment, the soil salinity in I3W1 was the lowest, followed by I2W1 and I1W2, and I1W1 had the highest value.

**Table 3.** Grain yields in different irrigation treatments.

| Years | Treatments | Number of Grains per Panicle | Seed Setting Rate/% | Thousand Grain Weight/g | Grain Yield/kg·hm$^{-2}$ |
|---|---|---|---|---|---|
| 2017 | I1W1 [1] | 74.92 [a][2] | 75.01 [a] | 23.45 [a] | 3860.44 [b] |
| | I2W1 | 70.00 [a] | 76.60 [a] | 23.11 [a] | 3942.59 [a][b] |
| | I3W1 | 69.84 [a] | 75.63 [a] | 22.94 [a] | 3491.66 [b] |
| | I1W2 | 71.15 [a] | 78.75 [a] | 23.22 [a] | 4421.05 [a] |

[1] I1W1 represents shallow–frequent irrigation with aquaculture wastewater; I2W1 represents shallow–wet irrigation with aquaculture wastewater; I3W1 represents flooding irrigation with aquaculture wastewater; I1W2 represents shallow–frequent irrigation with fresh water. [2] The different letters indicate significant differences among treatments at 5% according to Duncan's multiple range test.

**Table 4.** Soil salinity in different irrigation treatments.

| Treatments | I1W1 [1] | I2W1 | I3W1 | I1W2 |
|---|---|---|---|---|
| Soil Salinity/‰ | 2.34 | 2.24 | 1.98 | 2.25 |

[1] I1W1 represents shallow–frequent irrigation with aquaculture wastewater; I2W1 represents shallow–wet irrigation with aquaculture wastewater; I3W1 represents flooding irrigation with aquaculture wastewater; I1W2 represents shallow–frequent irrigation with fresh water.

## 4. Discussion

### 4.1. Water Quality Indexes

In this study, we analyzed the changes in DO, EC, pH, and ORP under aquaculture wastewater under different irrigation treatments.

#### 4.1.1. DO

Under natural conditions, the major sources of DO in rice fields were the dissolution of oxygen in the atmosphere and the oxygen secretion of rice roots (radial oxygen loss, ROL), and both changed the distribution of DO in rice fields [19]. The plants had a lower ratio of photosynthesis rate to respiration rate at night than in daytime [20]. Therefore, the dissolution of oxygen in the atmosphere was low at night, especially in the early morning [21]. In this experiment, water was sampled at 6:00, thus, the measured concentration of DO was generally low. As the water temperature increased, the amount of DO decreased in the water [22]. The decrease of DO at the late stage of tillering might be because of the high temperature in the mid-summer period. Dissolved oxygen is one of the limiting factors of the aquatic biological metabolism. Tim Rixen [23] demonstrated that wastewater discharge seemed to reduce oxygen concentration during the period of investigation. As shown in Figure 1, the concentration of DO was the highest in I1W1 but the lowest in I1W2 15 to 20 days after transplanting. That might be due to the fact that the shallow–frequent irrigation (I1) was replaced daily with aquaculture wastewater, thus DO escaped less in the surface water than it did in I2 and I3. As for the I1W2 treatment, the fresh water was replaced daily. The fresh water was pumped from a well. Below the water table, where gas exchange with the atmosphere ceased, DO was gradually consumed by microbial uptake, biodegradation of organic matter, and reaction with reduced mineral phases in the aquifer [24]. As shown in the Figure 1, during the late stage of tillering to the end of the jointing booting stage, the DO concentration of the I3W1 treatment was the highest in the three aquaculture wastewater irrigation treatments. This could be due to the formation of aerenchyma with the growth and development of rice roots. The stress of low oxygen concentrations in a waterlogged environment is minimized in some plants that produce aerenchyma, a tissue characterized by prominent intercellular spaces. It is produced by the predictable collapse of root cortex cells, indicating a programmed cell death (PCD) and facilitates gas diffusion between roots and the aerial environment. When cortex aerenchyma begins to form, flooding is beneficial to its formation. As for the I3W1 treatment, the duration of aquaculture wastewater kept in the soil surface was the longest, so the aerenchyma in I3W1 developed better. The aerenchyma could overcome waterlogging stress resulting in oxygen limitation in rice roots [25]. Changes in DO cycling patterns in surface waters have been shown to be useful as a potential indicator of biodegradable organic pollution. Dissolved oxygen may be useful as an indicator of biodegradable organic pollution load [26]. As shown in Figure 1, I3W1 was an optimal treatment and has value in creating a pattern of purification for aquaculture water.

#### 4.1.2. EC

Pan [27] indicated that the EC was related to electrolyte content and was one of the important indexes of water quality. The EC in water is influenced by ion species concentration dissolved in water and water temperature [28]. As shown in Figure 1, as a result of the salinity soil, high salt content in the rice soil in the early stage of rice transplanting contributed to the high concentration of EC in

surface discharge. The application of tiller fertilizer might lead to an increase in the EC value during the initial tillering period. As the fertilizer was absorbed by the rice and the fertilizer was lost in discharge treatments, the value of EC gradually reduced [29]. When irrigation and discharge increased, the salinity of the soil also decreased, and some fertilizer was also dissolved into the surface water and was discharged. Thus, the EC also decreased. The EC concentration steadily decreased from the tillering stage to the booting stage. The EC concentration of I3W1 was the lowest at the end of the jointing stage, which might be due to the long hydraulic retention time and large irrigation depth which has also been concluded by De-Feng [30].

### 4.1.3. pH

The dynamics of pH in the surface water of soil might be affected by agricultural practices (e.g., N fertilization, flooding regime, cropping system) and by different climatic scenarios (e.g., temperature and $(CO_2)$) [31]. The $CO_2$ in the water interacted with water, carbonate, and bicarbonate to form a complex reversible carbonate system. The $CO_2$ content in the surface water was influenced by photosynthesis of plants, respiration of aquatic organisms, water temperature, and oxidative decomposition of organic matter [32]. When the water contained decaying organisms or other organic pollutants, the organic matter could be oxidized into $CO_2$, $HNO_2$, and $HNO_3$, resulting in the decrease in pH [33]. As shown in Figure 1, the pH increased during the tillering stage. That might be because plants absorbed and utilized the organic fertilizer in aquaculture wastewater and fertilizer in soil with the rice plants growth and the concentration of decomposed organic matter in water was reduced. That leaded to the decrease in the content of $CO_2$ in surface water. As shown in Figure 1, the pH decreased during the jointing and booting stage. As shown in Figure 6, that might be due to the increasing of leaf areas. The leaves of different layers of plants were blocked by each other resulting in the decrease in photosynthetic effective radiation in the upper layer close to the surface water [34]. In addition, the dissolution of $CO_2$ in the atmosphere, which kept close to the surface water, was high and led to the decrease in pH in the surface water. The air was difficult to circulate; therefore, $CO_2$ was difficult to spread. There was little $O_2$ content in surface water, and anaerobic decomposition produced organic acids [35], which could be another reason for the pH decrease. The I1W1 and I1W2 were the shallow–frequent irrigation, and the duration of flooding was shorter than I2W1 and I3W1 (Table 1). Thus, the concentration of $CO_2$ dissolved in the surface water of I1W1 and I1W2 was less than that of I2W1 and I3W1. This was one of the reasons for the higher pH value in I1W1 and I1W2. The short duration of flooding led to less decomposition of organic matter and the content of $CO_2$ was less, which was released by oxidative decomposition of organic matter. This was another reason for the higher pH value in I1W1 and I1W2. As shown in Figure 1, the pH value in I2W1 was the lowest. The I2W1 was the shallow–wet irrigation, and the duration of flooding was longer than I1W1 and I1W2 (Table 1). Thus, the concentration of $CO_2$ dissolved in the surface water of I1W1 and I1W2 was less than that of I2W1. In addition, the duration of the flooding in I2W1 might have been the most active period of organic decomposition. During this time, a large amount of organic matter in wastewater was decomposed and a large amount of $CO_2$ was released. All of these could lead to the lower pH value in I2W1. The I3W1 was the flooding irrigation treatment, and the duration of flooding was the longest among all treatments (Table 1). Thus, the concentration of $CO_2$ dissolved in the surface water was the highest. But the duration of flooding might be too long and the amount of remaining organic matter was too small. The content of $CO_2$ in I3W1 was less than that in I2W1 which was released by oxidative decomposition of organic matter. Thus, the pH value in I3W1 was higher than that in I2W1.

### 4.1.4. ORP

Oxidation-reduction potential was a measure of the activity of electrons involved in oxidation-reduction reactions within an aqueous environment. As a tool for monitoring oxidative conditions, one of ORP's most important attributes was its utility over the full range of redox conditions, from highly reduced redox conditions to highly oxidized redox conditions [36]. As shown in Figure 1,

the value of ORP was the highest in I1W1 treatment and the lowest in I3W1 treatment during the jointing and booting stage. By comparison with Figure 1, the variation in the ORP value was similar to the variation in the pH value. The exact values of the ORP reflected all of the factors that contributed to the electron activity such as chemical constituents of the system, variety of biological activity, pH, and temperature. Oxidation-reduction potential was analogous to pH in many ways and real-time ORP response in different systems was like pH response in systems with varying buffer capacity [36]. Interestingly, no distinct correlation was found between DO and ORP and we could not explain this observation. While of interest, this topic is beyond the scope of the current paper's study and it should be noted that similar findings were previously found [37].

*4.2. The TN and TP Losses through Discharge*

In this study, we analyzed the changes in TN and TP losses for discharge under different irrigation treatments. As shown in Figures 2 and 3, the TN and TP concentrations and losses of surface discharge in paddy fields during the tillering stage were the highest in the three key stages. Because of the application of the basal fertilizer, regreening fertilizer and tillering fertilizer, the concentrations of TN and TP in the surface water were high. Frequent irrigation and discharge treatments resulted in a large losses of TN and TP, which gradually reduced the mass concentration of TN and TP in the surface water [38]. The concentration and losses of TP in surface discharge during the tillering stage were more significant than those during the jointing and booting and heading and flowering stages. It might be because the TP in aquaculture wastewater and fertilizer was easily adsorbed and fixed by the topsoil of the rice field, and the disturbance of the soil by irrigation and discharge treatments increased the release of TP from the topsoil into the surface water [39]. With the downward migration of TN and TP into the soil, the rice soil absorbed the nutrients in the irrigation water through the inorganic nitrogen absorption (nitrification-denitrification), the fixed deposition of phosphorus and the absorption of nutrients by the rice, so that the concentrations and losses of TN and TP in surface discharge gradually decreased [40]. As shown in Figures 4 and 5, the concentrations and losses of TN and TP in groundwater discharge were the highest during the tillering stage in the three key stages, which was similar to the variation in surface discharge. Because of the application of the basal, regreening and tillering fertilizer, the concentrations of TN and TP in the groundwater were high. It might also be affected by agronomy techniques such as rice transplanting, which made nitrogen leaching extremely at the early stage of rice growth [41]. Over time, the roots of the crops became mature and grew vigorously. In addition, the surface discharge lost a lot of TN, so that the concentrations and losses of groundwater discharge TN gradually decreased [42]. As shown in Figure 5, the TP concentration in groundwater discharge during the tillering stage was the highest in the three key growth stages, but there was no obvious differences in TP concentrations among the three stages, and the TP concentrations ranged from 0.054 mg/L to 0.065 mg/L. The losses of TP in groundwater discharges were low in all three key growth stages, which might be because of phosphorus retention mechanisms. Phosphorus retention mechanisms included uptake by plants, sorption, and exchange reactions with soils [43]. The concentration and total losses of TN in the I3W1 treatment were the lowest in the surface discharge during the three key stages when compared with the other three treatments. The main reason might be that the irrigation water of I3W1 was deeper and the hydraulic residence time was longer than those of other treatments. Yuyuan [17] concluded that the pollutant removal efficiencies under different hydraulic retention times were also significantly different. With the extension of hydraulic retention time, the decontamination effect was higher. Comparing with the other three treatments, the total losses of TP in the I3W1 treatment were the lowest in the surface discharge during the three key stages. As shown in Table 1, total discharge time and amount of I3W1 were the lowest of all treatments, therefore total losses of TP in the surface discharge was the lowest. As shown in Figures 2–5, the total losses of TN and TP in groundwater discharge were much lower than those in surface discharge and the total losses of TN and TP in surface and groundwater discharge of the I3W1 treatment were the lowest among all four treatments over the three key stages.

### 4.3. Growth Indexes

As shown in Figure 6, the growth indexes of the I1W1 treatment were always lower among all four treatments. The cause of this phenomenon needs further study. As shown in Figure 6, the plant height of I1W2 was the highest and that of I3W1 was lower than that of I1W2 and I3W1. Plant height is a trait modified by the environment [44], and long duration of flooding might reduce plant height [45]. As shown in Figure 6, the number of tillers for I2W1 were higher. The frequency of irrigation for I1W1 and I1W2 was too high to tiller well, meanwhile, the water depth of irrigation for I3W1 was too high to tiller well. As shown in Figure 6, the difference in stem diameter among the I1W1, I2W1, and I3W1 treatments was not significant after the jointing and booting stage, but that of the I1W2 treatment was significantly higher. The reason might be the differences among the two water sources. The increase in organic fertilizer input in aquaculture wastewater might not significantly promote the absorption of nitrogen, phosphorus, and potassium in crops and may have little effect on the stem diameter of rice [46]. As shown in Figure 6, the leaf area of I3W1 was the largest among the three aquaculture wastewater treatments. The main reason might be that the irrigation water of I3W1 was deeper and the duration of flooding was longer due to which the nitrogen fertilizer played a leading role in the leaf area of rice and the TN and TP in aquaculture wastewater, especially TN in I3W1, were better absorbed by plants [47].

### 4.4. Grain Yields

As shown in Table 3, the yield in I2W1 was the highest and that in I3W1 was the lowest among the three aquaculture wastewater treatments. As shown in Figure 6, the plant height of I2W1 was the highest among the three aquaculture wastewater treatments. Plant height is an important agronomic trait in rice, as it determines plant architecture and greatly influences the grain yield [48,49]. This yield increase might have been due to the higher tiller production. As shown in Figure 6, the number of tillers for I2W1 were the highest among the three aquaculture wastewater treatments. The yield in I3W1 was the lowest which might be due to the long duration of flooding [50].

### 4.5. Soil Salinity

Before the plant transplanting, the salt content of the paddy soil for all treatments was measured as 4‰. At the end of the experiment, the salt content of the paddy soil for all treatments decreased. The overall salt migration trend for all the soil salinity treatments was downward, which could be the result of frequent leaching [51]. As shown in Table 4, the desalinization rate of I3W1 was the largest, and that of I1W1 the lowest. When water depth and duration of flooding increased, the desalinization rate also improved. The result might be due to the higher irrigation water depth of I3W1, and longer hydraulic residence time; therefore, the amount of water infiltration was large and the salt removal effect was better [52].

### 4.6. Significance

The aim of this experiment was to investigate an optimal irrigation mode which could make full use of aquaculture wastewater to achieve aquaculture purification and soil desalinization. This study has practical significance for alleviating the pollution of wastewater in coastal areas and promoting soil desalinization.

## 5. Conclusions

In summary, the present study demonstrated that water indexes of surface discharge in paddy fields, TN and TP loss of discharge in paddy fields, growing indexes of plants, grain yield, as well as soil salinity were affected by different irrigation treatments and different irrigation water. However, it should be noted that, in this study, the focus was mainly on the responses of rice plants under different irrigation treatments with different irrigation water depths and different hydraulic residence

times. We analyzed the variation of water indexes of surface discharge in paddy fields, and the results revealed that the values of DO, EC, pH, and ORP were optimal in I3W1; collectively, the total losses of TN and TP of surface and ground water discharges of the I3W1 treatment were the lowest among all four treatments over the three important stages; the gain yield of I2W1 was the highest among the aquaculture irrigation treatments; and the desalinization rate of I3W1 was the largest. In general, the I3W1 treatment was the optimal treatment. To better clarify the mechanism of advantage for the different irrigation treatments in soil salinity, it is necessary to do the experiment with deeper water depths and longer hydraulic residence times, respectively, in a further study.

**Author Contributions:** Conceptualization, Z.W.; methodology, Y.X. and S.L.; validation, Z.W. and X.G.; formal analysis, Y.X. and S.C.; data curation, Z.X. and Y.A.H.; supervision, Z.W. and X.G.; project administration, Z.W. and X.G.; writing—original draft preparation, Y.X.; writing—review and editing, Y.X., Z.W., X.G., S.L., C.S., Z.X. and Y.A.H.

**Funding:** This research was funded by the Fundamental Research Funds for the Central Universities (2019B18114) and by the National Natural Science Foundation of China (51309080) and Water Conservancy Science and Technology Project of Jiangsu Province.

**Acknowledgments:** The authors also thank the reviewers and editors for their valuable comments about the manuscript.

**Conflicts of Interest:** The authors declare no conflict of interest.

## Abbreviations and Symbols

| | |
|---|---|
| TN | total nitrogen |
| TP | total phosphorus |
| I1 | shallow–frequent irrigation |
| I2 | shallow–wet irrigation |
| I3 | flooding irrigation |
| W1 | aquaculture wastewater |
| W2 | fresh water |
| CF | compound fertilizer |
| U | urea |
| DO | dissolved oxygen |
| EC | electrical conductivity |
| pH | potential of hydrogen |
| ORP | oxidation-reduction potential |

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
