# Peer review of "Effects of Different Irrigation Treatments on Aquaculture Purification and Soil Desalination of Paddy Fields"

_water, doi:10.3390/w11071424_

Round 1
Reviewer 1 Report
The manuscript is well written. Points are sound and clear. I have some minor questions to discuss. please find my comments in the attached file.

Author Response
Dear Reviewer,
We are grateful for your comments concerning our manuscript entitled “Effects of different irrigation treatments on aquaculture purification and soil desalination of paddy fields” (ID: water-517111). We sincerely thank the reviewers for constructive criticisms and valuable comments. The comments are encouraging, and the reviewers appear to share our judgment that this study and its results are important to aquaculture purification and soil desalination of paddy fields. The main corrections are highlighted and included in the revised manuscript. Corrections relative to English language are displayed within Track Changes mode. Our detailed responses to the comments are given in the revised manuscript and the attached report.
Yours sincerely
Yi Xie

Reviewer 2 Report
Generally the English is fine. There are some minor errors e.g. Line 32 introduction worlds should read world. Please recheck the English when the MS is revised.
Place a list of abbreviations and symbols used before the introduction or after the conclusions.
The methodology used is clear and the approach used could be reproduced by another researcher.
Line 77-80 - Please define how the concentrations were measured.
Line 134 - Please define the instruments used for DO, EC, pH and ORP to gether with the calibration approach used, if any
Line 165 - 388 reader access would be improved if you used additional subheadings to break up the text and direct the reader to the point you are making
Line 388 - 547. Please divide this section into a number of subsections to improve reader access
Line 548 - Please add a new section demonstrating how your study move the science forward and explaining why others should use your approach
Technically the study is OK. The text is clear, but could be improved by adding subheadings, with no more than two paragraphs in each subsection in the discussion and results sections. This change is requested to improve reader access to your MS. The tables are clear, the figures are clear, but the histogram figures 3-5 would be clearer if each data histogram within a figure was in a separate block colour.
Author Response

(The authors gave the same response as above.)
